# FAPi PET/CT Imaging to Identify Fibrosis in Immune-Mediated Inflammatory Diseases

**DOI:** 10.3390/biomedicines13040775

**Published:** 2025-03-22

**Authors:** Dalia A. Lartey, Lynn A. Schilder, Gerben J. C. Zwezerijnen, Geert R. A. M. D’Haens, Joep Grootjans, Mark Löwenberg

**Affiliations:** 1Laboratory of Experimental Oncology and Radiobiology, Cancer Center Amsterdam, Amsterdam University Medical Centers, 1081 BT Amsterdam, The Netherlands; d.a.lartey@amsterdamumc.nl (D.A.L.);; 2Department of Gastroenterology and Hepatology, Amsterdam University Medical Centers, 1081 HV Amsterdam, The Netherlands; 3Oncode Institute, 3521 AL Utrecht, The Netherlands; 4Department of Radiology and Nuclear Medicine, Amsterdam University Medical Centers, 1081 HV Amsterdam, The Netherlands

**Keywords:** fibrosis, immune-mediated inflammatory diseases (IMIDs), FAPi PET/CT, fibroblast activation protein (FAP), positron emission tomography imaging

## Abstract

Immune-mediated inflammatory diseases (IMIDs) are characterized by chronic systemic inflammation and multi-organ involvement. Fibrosis formation in IMIDs can cause tissue destruction and subsequently organ malfunction. Fibroblast activation protein inhibitor positron emission tomography/computed tomography (FAPi PET/CT) represents a novel imaging technique that holds great potential to visualize in vivo fibrosis. We here provide an overview of available evidence on FAPi PET/CT imaging to visualize fibrosis in various IMIDs, including interstitial lung diseases, immunoglobulin G4-related diseases, cardiovascular diseases, kidney diseases, and gastrointestinal diseases. FAPi PET/CT imaging demonstrates high sensitivity in detecting early fibrosis, correlating with disease severity, across different IMIDs, showing superiority compared to conventional imaging modalities. Although FAPi PET/CT might be a useful tool to assess fibrosis formation, thereby aiding in grading disease severity and staging, future studies should include larger sample sizes in a broad variety of IMIDs with emphasis on the optimization of imaging protocols to further validate its diagnostic value.

## 1. Introduction

Immune-mediated inflammatory diseases (IMIDs), such as inflammatory bowel disease (IBD), systemic sclerosis, spondylarthritis, and interstitial lung diseases, are characterized by chronic systemic inflammation frequently leading to fibrosis formation in various organs [1]. Although the exact underlying mechanisms remain to be defined, it is generally hypothesized that the development of fibrosis is primarily caused by longstanding tissue injury, myofibroblast activation and expansion of mesenchymal stroma cells (MSCs) in synovial membranes or parenchymatous organs. These activated myofibroblasts represent a mesenchymal cell subset that regulate accumulation and degradation of the extracellular matrix. In the process of fibrogenesis, there is an overshoot of extracellular matrix accumulation compared to degradation [2].

Fibroblast activation protein alpha (FAP) is a transmembrane glycoprotein with dipeptidyl peptidase and endopeptidase activity, which, as its name suggest, is primarily expressed by activated fibroblasts [3]. These activated fibroblasts play a crucial role in various pathophysiological conditions, including tumor growth, wound-healing, migration of tumor cells, inflammation, and fibrogenesis [4,5]. FAP levels have been shown to be elevated in IMIDs due to chronic inflammation, tissue response, and fibrosis [3]. Positron emission tomography/computed tomography (PET/CT) with a FAP inhibitor (FAPi) labeled with radioactive isotopes fluorine-18 (18F) or gallium-68 (68Ga) represents a promising molecular imaging technique, with high sensitivity and specificity for the detection of inflammatory and fibrotic lesions [6]. This imaging modality allows visualization of in vivo FAP expression, indicating tissue remodeling, inflammation, and fibrosis in patients with IMIDs [1,5,7,8,9]. As the global availability and interest in FAPi PET/CT expands, the potential indications for its use are expected to increase. Therefore, the objective of this review is to evaluate FAPi PET/CT imaging in identifying fibrosis in various IMIDs, while providing an overview of both pre-clinical and clinical findings on FAPi PET/CT imaging to detect fibrotic lesions in IMIDs.

## 2. Pulmonary Diseases

Interstitial lung diseases (ILDs) consist of various diseases, and they are known to have a poor prognosis in a substantial proportion of patients, as they are associated with irreversible pulmonary fibrosis [10]. Pulmonary imaging is essential for the diagnosis and follow-up of ILDs. High-resolution (HR) CT imaging and conventional imaging with Fluoro-D-glucose (FDG)-PET are the most frequently used imaging modalities. Pulmonary function tests are also used to monitor lung function [11]. The potential of FAPi PET/CT imaging to visualize in vivo fibrosis in ILD was reported by several studies, as shown in Table 1. For instance, Bahtouee et al. suggested that 68Ga-FAPi-46 PET imaging could serve as a non-invasive method for evaluating active fibrosis and inflammation in ILD patients [12]. Additionally, imaging with Technetium-99m sestamibi scintigraphy was reported to increase sensitivity and specificity for fibrosis evaluation while also differentiating levels of fibrotic activity based on high-resolution computed tomography patterns (HRCT) [12]. These findings were further validated by Röhrich et al., who showed increased FAPi uptake in patients with fibrotic ILD (fILD) [13]. Additional dynamic imaging analysis illustrated a difference between FAPi uptake in fILD patients compared to ILD patients with lung cancer. The so-called tissue to background ratio increased over time in lung cancer patients, while remaining stable in fILD. Dynamic PET imaging revealed fILD lesions with an early peak, followed by a slow decrease in signal intensity. Lung cancer patients had a late peak, followed by a gradual washout phase. Therefore, dynamic FAPi PET imaging was suggested to serve as a valuable imaging modality to distinguish benign from malignant lesions [13]. Furthermore, Mori et al. assessed the potential of 18F-FAPi-74 PET imaging in idiopathic pulmonary fibrosis (IPF) compared to controls [14]. FAPi-positive areas were seen in fibrotic lesions in the lung parenchyma. Significantly higher levels of the maximum and mean standardized uptake value (SUVmax and SUVmean) were found in fibrotic lung tissue as compared to the control group. Moreover, measured fibrotic active volume of IPF patients could delineate fibrotic changes and a strong correlation between FAPi imaging and HRCT was reported (R = 0.887, *p* < 0.005). Overall, these data suggest that FAPi can serve as a non-invasive imaging technique to detect and assess pulmonary fibrotic processes and changes in IPF. Additionally, FAPi was able to predict disease progression by quantifying fibrosis, with tracer uptake being higher in IPF patients compared to controls [14]. Another study showed that FAPi PET imaging was a promising imaging modality to detect pulmonary fibrosis in mice [15]. Bleomycin treated and control mice underwent 68Ga-FAPi-46 PET/CT imaging at 7 and 14 days post-treatment. On day 14, increased FAPi uptake was seen in the bleomycin group compared to controls. This was further supported by histological and immunohistochemical analyses, where higher collagen deposition and FAP expression was seen in lung samples acquired from both the bleomycin and control group [15]. These findings were supported by recent data from Ji et al., who investigated the possibility of 68Ga-FAPi PET/CT imaging compared with 18F-FDG PET/CT imaging to detect pulmonary fibrosis in mice with bleomycin-induced pulmonary fibrosis. FAPi imaging was able to detect fibrosis formation in these mice. Upon treatment with pirfenidone, an approved anti-fibrotic drug for IPF, both FDG and FAPi uptake showed a significant decrease. This suggests that FAPi imaging also holds promise to monitor treatment responses in fibrotic diseases [16]. Interestingly, a recent study by Chen et al. suggested that imaging with a heterobivalent probe, which visualizes not only FAPi-46 but also Dotatate-LM3 (DOTA-LM3), a somatostatin receptor (SSTR) antagonist, may open new doors in early fibrosis detection [17]. In this pre-clinical study, mice with bleomycin-induced fibrosis underwent 68Ga-FAPI-46, 68Ga-DOTA-LM3, and 68Ga-FAPI-LM3 PET imaging on day 7 post-induction. Increased tracer uptake in mice with early pulmonary fibrosis using combined imaging with FAPi and LM3 was significantly higher compared to monomeric tracers. These data were further corroborated by increased mRNA and protein expression levels of FAP and somatostatin receptor 2 in pulmonary samples obtained from these mice [17].

## 3. IgG4-Related Diseases

Immunoglobulin G4 (IgG4)-related diseases are multi-organ diseases which are characterized by local and/or systemic inflammation and fibrosis formation. Inflammation and fibrosis are typically present in salivary and lacrimal glands, retroperitoneal organs, and lymph nodes, in these patients [18]. FDG PET/CT is often used to assess the severity of organ involvement and monitor disease progression [19]. Schmidkonz et al. aimed to differentiate inflammatory from fibrotic disease activity in fibroinflammatory conditions, by comparing 68Ga-FAPi-04 PET/CT with 18F-FDG-PET/CT (Table 2 and Table 3) [20]. Results showed FDG uptake in inflammatory and mixed fibroinflammatory phenotypes, whereas fibrotic (i.e., without signs of active inflammation) lesions remained FDG-negative. In contrast, inflammatory manifestations were negative for FAPi, whereas fibrotic lesion showed clear uptake. FDG and FAPi uptake showed no significant correlation, which might be due to various gradations in both inflammation and fibrosis. Thus, FAPi and FDG PET/CT seem to be suitable for the investigation of different stages of IgG4-related diseases as a fibro-inflammatory disease, especially in distinguishing between inflammatory and non-inflammatory fibrosis. Follow-up imaging did not show reduced FAPi uptake after anti-inflammatory treatment compared to baseline, while FDG uptake was significantly reduced. FAPi and FDG PET/CT were also compared by Luo et al. [21]. Findings demonstrated more pronounced FAPi uptake levels in the pancreas, lacrimal gland, sublingual gland, submandibular gland, bile ducts and liver, compared with FDG. Nevertheless, FAPi activity was absent in affected lymph nodes that were FDG-positive. Moreover, the low background uptake of FAPi in the head and neck provides an advantage over FDG for detecting IgG4-related diseases in these organs [21].

## 4. Cardiovascular Diseases

Systemic amyloid light chain (AL) can involve various organs, such as the kidney and heart. AL cardiac amyloidosis (CA) is often underdiagnosed or misdiagnosed [27]. Thus, early and accurate detection of AL CA is essential. Wang et al. aimed to evaluate the feasibility of 68Ga-FAPi-04 PET/CT in assessing AL CA [23]. Patients underwent FAPi PET/CT, cardiac magnetic resonance (CMR), echocardiography, and clinical parameters such as N-terminal pro-brain natriuretic peptide and cardiac troponin I were assessed. The patients were subsequently categorized into three groups based on uptake patterns of the tracer, i.e., patchy (PET-patchy), extensive (PET-extensive), and negative (PET-negative). The SUV ratio and left ventricle molecular volume were significantly higher in the PET-extensive group compared to the PET-patchy group (*p* = 0.005). In addition, FAPi-uptake showed a significant correlation with clinical parameters, such as heart structure and function metrics. This suggested that FAPi PET/CT could assist in identifying light-chain cardiac amyloidosis by detecting myocardial fibroblast activation associated with myocardial remodeling, and also demonstrated a strong correlation with disease severity [23].

Large vessel vasculitis (LVV) is a rare disease, which can lead to active inflammation and chronic vessel damage [28]. Röhrich et al. aimed to detect fibroblast activation in vessel walls with 68Ga-FAPi-46 in active and inactive LVV [22]. FAPi PET/CT imaging, CT and magnetic resonance imaging (MRI) scans were performed. FAPi uptake was detected in all patients with LVV and was absent in the control group. One patient with active LVV demonstrated the highest FAPi uptake. Interestingly, patients who were in long-standing remission also showed FAPi activity. For instance, significantly increased FAPi tracer uptake was observed in the aortic arch of a patient with inactive LVV. Thus, FAPi uptake was significantly higher in LVV compared to controls and fibroblast activation in vessel walls in LVV could be visualized using FAPi PET/CT [22].

## 5. Renal Diseases

Progression in chronic kidney diseases often leads to renal fibrosis, which is considered to be irreversible. Renal fibrosis is a key predictor of a patient’s prognosis [29]. Therefore, accurate assessment of renal fibrosis is necessary. Zhou et al. studied patients who underwent kidney biopsies to determine renal fibrosis [25]. 68Ga-FAPi-04 PET/CT and multiple staining techniques were performed to assess glomerular fibrosis, renal interstitial fibrosis and disease severity (graded as: mild I, moderate II, or severe fibrosis III), compared to controls. Out of the thirteen included patients, four had renal fibrosis caused by an immune-mediated condition, including membranoproliferative glomerulonephritis (MPGN) and crescentic glomerulonephritis. All four patients had moderate or severe fibrosis with increased SUVs compared to the control group. FAPi PET/CT was considered a promising imaging technique to detect renal fibrosis, which can aid in assessing disease progression, diagnosis, and management [25].

IgAN is a subtype of glomerulonephritis and a leading cause of kidney failure worldwide, resulting in progressive inflammation and eventually fibrosis [30]. Wang et al. aimed to evaluate the severity of IgAN with 18F-AlF-NOTA-FAPi-04 PET/CT imaging [24]. Kidney biopsy tissues were analyzed using immunofluorescent staining. Additionally, immunohistochemical staining was performed to evaluate FAP expression. A significant negative correlation was found between estimated glomerular filtration rate levels and SUVs in the renal parenchyma (*p* < 0.001). The SUVmax and SUVmean in this patient group were significantly higher than those in the control group (*p* < 0.001). Moreover, analysis of interstitial fibrosis demonstrated lower SUVmax and SUVmean levels observed in lesions with mild interstitial fibrosis and tubular atrophy, in contrast to those with moderate or severe interstitial fibrosis and tubular atrophy. Based on this work, it was concluded that FAPi PET/CT can serve as a non-invasive technique for evaluating renal inflammation and fibrosis, providing valuable insights into the pathological severity in IgAN patients [24].

## 6. Gastro-Intestinal and Hepatic Diseases

Liver fibrosis is often considered a preliminary stage of irreversible liver cirrhosis. Hepatic fibrogenesis had been described to be predominantly mediated by hepatic stellate cells (HSCs) [31]. Continuous events of inflammation and tissue injury can lead to HSC activation and subsequent collagen deposition, eventually leading to fibrosis formation [32]. A recent (pre)clinical study by Song et al. investigated the potential of 68Ga-DOTA-FAPI-04 PET to identify early hepatic fibrosis in mice and patients with suspected or confirmed fibrosis [26]. Mice with CCl4-induced liver fibrosis and controls underwent FAPi PET imaging at various timepoints. Immunofluorescence, immunohistochemical, and histological analyses were performed to assess FAP and alpha smooth muscle actin protein expression, as well as hepatic disease activity. Higher tracer uptake was seen in mice with active fibrosis as compared to controls. Furthermore, histological analyses showed increased fibrotic markers in hepatic tissue of CCL4 treated mice compared to controls. For the clinical part of the study, 26 patients with suspicion of liver fibrosis underwent FAPi PET imaging with CT or MRI. Additionally, serum biomarkers (including alanine aminotransferase and aspartate aminotransferase) were acquired from participants. FAPi uptake was significantly correlated to inflammation and fibrosis. Interestingly, the SUV showed distinguishing capacities between inflammation and fibrosis, further strengthening the potential of FAPi PET imaging to detect early stages of liver fibrosis [26].

Crohn’s disease (CD) is a subtype of IBD, known for its inflammatory and fibrotic manifestations in the gastrointestinal tract [7]. Progressive CD often leads to complications such as strictures, perforations, fistulas, and abscess formation, which often require surgical interventions [33]. In order to prevent disease progression, adequate disease monitoring is necessary. Chen et al. assessed multiple intestinal segments of CD patients using CT enterography (CTE), 68Ga-FAPi-04 PET/CT, and ileocolonoscopy (Table 3 and Table 4) [34]. In total, 74 intestinal segments were investigated. Out of these 74 segments, 45 were identified by ileoncolonoscopy, of which FAPi PET/CT identified 42 segments with a sensitivity of 94% and specificity of 72%. CTE identified 39 out of 45 segments with a sensitivity of 87% and a specificity of 45%. Furthermore, FAPi detected more mild to moderate lesions than CTE. Additionally, some segments were positive for both Ga-FAPi and CTE despite the absence of endoscopic lesions, potentially indicating submucosal fibrosis. Thus, these data indicate that FAPi PET/CT might be superior in detecting endoscopic lesions compared to CTE, indicating the value of FAPi imaging to assess disease severity in CD. Of note, fibrosis is a multi-stage process, and FAPi uptake might therefore differ according to the severity of fibrosis formation. For instance, a study by Scharitzer et al. reported differences in SUVmax between various degrees of fibrosis, with more severe fibrosis having a higher maximum SUVmax. Although this study used PET/MRI and not PET/CT, tracer uptake was shown to be higher in more severe fibrotic lesions [35].

## 7. Conclusions

Pre-clinical and clinical studies suggest that FAPi PET/CT represents a novel advanced imaging technique to visualize fibrosis in IMIDs, with superior specificity and sensitivity compared to conventional imaging methods, providing important insights into the pathophysiology of fibrosis. Despite its potentially great value in the early detection of fibrosis, many variables such as the most optimal FAPi tracer and adequate acquisition protocol including uptake time, single- versus multi-time point imaging or (short) dynamic imaging for FAPi PET/CT imaging are yet to be fully elucidated. Furthermore, reliable qualitative and (semi)quantitative uptake metrics are yet to be addressed as in-depth knowledge about pharmacokinetics of FAPi tracers in the setting of inflammatory fibrosis is still limited, questioning the reliability of the simplistic SUV metrics (as for FDG PET) for disease assessment, despite being dominantly applied in FAPi PET imaging studies which assess fibrosis. To address these matters, future studies should include larger sample sizes in a broad variety of IMIDs with emphasis on the acquisition protocol in order to further validate its diagnostic value. Investigating the utility in monitoring treatment responses and combining FAPi PET/CT with fibrosis and inflammatory biomarkers and with conventional imaging techniques will likely enhance its clinical application. FAPi PET/CT imaging might facilitate early fibrosis detection, thereby improving diagnostic, monitoring, and treatment stratification of fibrosis in IMIDs.

## Figures and Tables

**Table 1 biomedicines-13-00775-t001:** FAPi PET/CT in pulmonary diseases.

Author, Year	Country	Study Design	Imid	Study Population (% Male)	Control Group (n)	Age (Years)	Comparison with 18F-FDG	Additional Parameters	Main Results
CHEN et al., 2023 * [17]	China	Prospective	Pulmonary fibrosis	3 mice, exact number not specified	Yes	N/A	No	Immunofluorescence, Western blotting, RT-qPCR.	68Ga-FAPi-LM3 which targets not only FAP, but also SSTR2, is a promising potential tracer for early diagnosis of pulmonary fibrosis.
JI et al., 2024 * [16]	China	Prospective	Pulmonary fibrosis	19 mice	yes (n = 7)	N/A	Yes	Histological staining and FAP and GLUT1 immunohistochemical staining.	FAPi-PET can assess fibrosis and treatment response in a pulmonary fibrosis mouse model. FAP expression corroborates this.
ROSENKRANS et al., 2022 * [15]	United states of America	Prospective	Pulmonary fibrosis	3–5 mice per subgroup, exact number not specified	Yes (not specified)	N/A	Yes	Masson’s trichrome staining and FAP immunohistochemical staining.	68Ga-FAPi-46 lung uptake was significantly increased n bleomycin mice compared to controls at both day 7 and 14. 18F-FDG uptake was highest during the inflammatory phase of the mouse model (day 6) and a non-significant decrease was seen during the profibrotic phase.
BAHTOUEE et al., 2024 [12]	Iran	Prospective	Interstitial Lung Disease	20 patients (40%)	Yes (n = 10)	Mean: 58.7	No	HRCT fibrosis score, ESR, CRP tests and pulmonary function tests.	68Ga-FAPi PET/CT, combined with 99mTc-MIBI SPECT/CT, could serve as a noninvasive method for evaluating active fibrosis and inflammation in ILD patients, while also differentiating levels of fibrotic activity based on HRCT patterns.
MORI et al., 2024 [14]	Chile	Prospective	Idiopathic pulmonary fibrosis	8 patients (50%)	Yes (n = 6)	Median: 71	No	HRCT-scan (HUmean and Humax) FVC, DLCO.	18F-FAPi-74 PET/CT could serve as a non-invasive imaging technique to detect and assess pulmonary fibrotic processes and changes in IPF. Additionally, FAPi could potentially predict disease progression by quantifying fibrosis, with tracer uptake being higher in IPF patients compared to controls.
RÖHRICH et al., 2022 [13]	Germany	Retrospective	Fibrotic Interstitial Lung Diseases	15 patients -	No	Mean: 71.2	No	CT-based fibrosis scores (FIB-index and GGO-index) FAP immunohistochemical staining of fILD biopsies.	FILD lesions showed an elevated 68Ga-FAPi uptake, thus FAPi could be a valuable tool in managing fILD, especially in predicting disease progression and evaluating therapy response. Also, immunohistochemistry of human biopsy samples showed a patchy expression of FAP in fibrotic lesions, preferentially in the transition zone to healthy lung parenchyma.

* (partly) pre-clinical study. CRP: C-reactive protein; DLCO: diffusing capacity for carbon monoxide; ESR: erythrocyte sedimentation rate; FAP: fibroblast activation protein; FAPi: fibroblast activation protein inhibitor; FDG: fluorodeoxyglucose; FIB: fibrosis index; fILD: fibrotic interstitial lung disease; FVC: forced vital capacity; GGO-Index: ground-glass opacity index; HRCT: high-resolution computed tomography; HU: Hounsfield unit; IPF: idiopathic pulmonary fibrosis; LM3: p-Cl-Phe-cyclo(d-Cys-Tyr-d-4-amino-Phe(carbamoyl)-Lys-Thr-Cys)d-Tyr-NH_2_; PET/CT: positron emission tomography/computed tomography; RT-qPCR: real-time quantitative PCR; SPECT: single-photon emission computed tomography; SSTR2: somatostatin receptor 2; 99mTc-MIBI: Technetium 99m sestamibi.

**Table 2 biomedicines-13-00775-t002:** FAPi PET/CT in cardiovascular, renal and IgG4-related diseases.

Author, Year	Country	Study Design	IMID	Study Population (% Male)	Control Group (n)	Age (Years)	Comparison with 18F-FDG	Additional Parameters	Main Results
LUO et al., 2021 [21]	China	Prospective	IgG4-related disease	26 patients (76.9%)	No	Mean: 51.5	Yes	CT-scan.	68Ga-FAPi PET/CT demonstrated a higher sensitivity accompanied with greater uptake in the bile duct, liver, salivary and lacrimal gland and pancreas compared to 18F-FDG in IgG4-RD. 68Ga-FAPi is not suitable for detecting IgG4-related lymphadenopathy.
RÖHRICH et al., 2024 [22]	Germany	Retrospective	LVV (Aortitis)	8 patients (12.5%)	Yes (n = 8)	Active LVV (n = 3) median: 58 Inactive LVV (n = 5) median: 59	No	MRI (inflammatory activity score), CRP.	FAPi uptake can be detected in aortitis patients even during remission, years after onset, and with low MRI inflammatory activity scores. This suggests that FAPI is a promising tracer for large vessel vasculitis (LVV) and may detect ongoing pathology in the vessel walls despite clinical remission.
SCHMIDKONZ et al., 2020 [20]	Germany	Prospective	IgG4 -related diseases	27 patients (70.4%)	No	Mean 54.9	Yes	Histological analysis and fluorescence imaging of the corresponding biopsies. RNA-sequencing of fibrotic and resting fibroblasts.	FAPi PET/CT enables differentiation between inflammatory and fibrotic activity in IgG4-related disease in vivo, with fibrotic activity indicated by increased FAPi uptake.
WANG et al., 2022 [23]	China	Prospective	Light-chain cardiac amyloidosis	30 patients (66.7%)	No	Mean: 59.1	No	CMR, clinical biomarkers and echocardiography.	FAPi PET/CT can assist in identifying light-chain cardiac amyloidosis by detecting myocardial fibroblast activation associated with myocardial remodeling, while also demonstrating a strong correlation with disease severity.
WANG et al., 2024 [24]	China	Prospective	Immunoglobulin A nephropathy	20 patients (65.0%)	Yes (n = 10)	Mean: 44	No	Renal biopsy, assessed using the Oxford classification, Masson trichrome staining for tubular atrophy and interstitial fibrosis, and immunohistochemical staining for FAP and αSMA.	18F-AlF-NOTA-FAPi-04 PET/CT can serve as a non-invasive technique for evaluating inflammation and fibrosis, providing valuable insights into the pathological severity in IgAN patients.
ZHOU et al., 2021 [25]	China	Retrospective	Renal fibrosis	13 patients (61.5%)	Yes (n = 9)	Mean: 42	No	Immunochemical analyses assessing glomerular fibrosis, renal interstitial fibrosis and the extent of disease.	68Ga-FAPi-04 is eligible to detect moderate-to-severe renal fibrosis caused by different forms of diseases.

αSMA: α-smooth muscle actin; CMR: cardiac magnetic resonance; CRP: C-reactive protein; FAP: fibroblast activation protein; FAPi: fibroblast activation protein inhibitor; MRI: magnetic resonance imaging; IgAN: immunoglobulin A nephropathy; PET/CT: positron emission tomography/computed tomography.

**Table 3 biomedicines-13-00775-t003:** Summary of main characteristics investigated in FAPi PET/CT.

Author and Year	Tracer	PET/CT Scanner	Administered Dose	Post Injection Imaging (min)	(Semi)Quantitative Uptake Metrics
**PRE-CLINICAL AND TRANSLATIONAL STUDIES**					
Chen et al., 2024 [17]	68Ga-FAPi-46, 68Ga-DOTA-LM3 and 68Ga-FAPi-LM3	Inveon small-animal scanner (Siemens)	7.4 MBq	30 + 5 and 60 + 5 min	%ID/g, VOI
Ji et al., 2024 [16]	68Ga-FAPI	Small-animal PET/CT scanner (Novel Medical)	6.0 MBq	50 min	HU, %ID/g, correlation %ID/g with percentage positive area of FAP immunohistochemical staining
Li et al., 2024 [8]	18F-FAPi	IRIS PET/CT system (Inviscan)	3.7 MBq	60 min	SUVmean, SUVmax, MTR
		Discovery MI (GE Healthcare)	1.8–2.2 MBq/kg		
Rosenkrans et al., 2022 [15]	68Ga FAPi-46 and 68Ga-DOTA	Inveon small-animal scanner (Siemens)	0.8–3.8 MBq for dynamic imaging and 3.8 MBq for static imaging	0 and 60 min	HU, %IA, %IA/cc, VOI
Song et al., 2024 [26]	68Ga-DOTA-FAPI-04	InliView-3000B and NMSoft-AIAC (NovelMedical™)	3.70–7.40 MBq	40 min	%ID/cc, SUVmax, ASUVmax, SUVmean, ASUVmax/B and ASUVmean/B.
		TOF-PET/MR (SIGNA, GE Healthcare) or Discovery VCT® (GE Healthcare)	100–218 MBq		
**CLINICAL STUDIES**					
Bahtouee et al., 2024 [12]	68Ga-FAPi-46	General Electric (Discovery IQ)	150 MBq	60 min	TL-FAPI, SUVmax, SUVmean, TLRmax, TLRmean, HUmax and HUmean
Chen et al. 2023 [17]	68Ga-FAPi-04	Umi 780	1.85–2.96 MBq/kg	60 ± 10 min	SUVmax, TBR, global FAPi PET/CT score
Luo et al., 2021 [21]	68Ga-FAPi	Polestar m660 (SinoUnion) or Biograph 64 TruePoint TrueV (Siemens)	85.2 ± 27.0 MBq	54.4 ± 15.8 min	SUVmax
Mori et al., 2024 [14]	18F-FAPi-74	Biograph mCT Flow scanner (Siemens)	199–239 MBq	60 min	SUVmean, SUVmax, FAV
Röhrich et al., 2022 [13]	68Ga-FAPi-46	Biograph mCT Flow (Siemens)	150–250 MBq	10 * and 60 and 180 min	SUVmean, SUVmax, TBR/LBR
Röhrich et al., 2024 [22]	68Ga FAPi-46	FlowMotion (Siemens)	177–285 MBq	60 min	SUVmean, SUVmax TBR, FAPi visual activity score
Schmidkonz et al., 2020 [20]	68Ga-FAPi-04	Biograph mCT 40 (Siemens)			SUVmax, SUVmean, MAV, TL-FAPi
Wang et al., 2022 [23]	68Ga-FAPi-04	Polestar m660 (SinoUnion)	107.4 ± 26.5 MBq	60 min	SUVmax, SUVmean, SUV ratio, LV molecular volume
Wang et al., 2024 [24]	18F-AlF-NOTA-FAPi-04	Biograph mCT Flow 64 (Siemens)	3.70–4.44 MBq/kg	61.2 ± 8.5 min	SUVmax, SUVmean
Zhou et al., 2021 [25]	68Ga-FAPi-04	Not specified	1.85–2.59 MBq/kg	50 to 60 min	SUVmax of kidney, SUVmean of liver and TBR

* Dynamic scans were conducted over a 40 min period instead of imaging after 10 min in three out of the eight patients, static imaging followed, after 60 and 180 min; %IA: percent injected activity; %IA/cc: percent injected activity per cubic centimeter of tissue; %ID/g: percentage injected dose per gram; %ID/cc: percentage injected dose per cubic centimeter; ASUVmax/B: ratio of average maximum standardized uptake volume to the background; ASUVmean/B: ratio of average mean standardized uptake value to the background; FAPi: fibroblast activation protein inhibitor; FAV: fibrotic active volume; HU: Hounsfield unit; LBR: lesion to back ratio; LV: left ventricle; MAV: metabolism activation volume; SUV: standardized uptake value; TBR: target to back ratio; TL: total lesion; VOI: volume of interest.

**Table 4 biomedicines-13-00775-t004:** FAPi PET/CT in gastro-intestinal and hepatic diseases.

Author, Year	Country	Study Design	IMID	Study Population (% Male)	Control Group (n)	Age (Years)	Comparison with 18F-FDG	Additional Parameters	Main Results
LI et al., 2024 * [36]	China	Prospective	Intestinal fibrosis, Crohn’s disease	22 rats	Yes (n = 4)	N/A	Yes	MTI, histopathology, FAP immunohistochemical staining of intestinal specimens.	FAPi-PET imaging was reported to hold great accurate diagnostic value in the early detection of intestinal fibrosis. In rats, FAPi uptake showed to have a higher correlation with fibrosis compared to FDG and MTI, specifically in early phase fibrosis. In patients with Crohn’s disease, FAPi superior than FDG in distinguishing fibrosis gradations.
SONG et al., 2024 * [26]	China	Prospective	Liver fibrosis	34 mice 26 patients (38.5%)	Yes (n = 26) No	N/A Mean: 50	No	Immunofluorescence and immunohistochemical analysis for FAP and αSMA expression. Histological analysis, serum markers, Forns index for inflammation and fibrosis, fibroscan.	68Ga-DOTA-FAPi-04 PET holds great potential to detect multiple stages of hepatic fibrosis. FAPi imaging was shown to be superior to other conventional markers in the early detection of fibrosis.
CHEN et al., 2023 [34]	China	Retrospective	Crohn’s disease	16 patients (68.8%)	No	Median: 23	No	Ileocolonoscopy (CDEIS and SES-CD) and CTE procedure, CDAI, CRP, FCP, global ileocolonic PET/CT score.	68Ga-FAPi-04 PET/CT demonstrated promising sensitivity and specificity for detecting endoscopic (fibrotic) lesions, with the severity of these lesions significantly correlating with the intensity of FAPi uptake.

* (partly) pre-clinical study. αSMA: α-smooth muscle actin; CDAI: Crohn’s disease activity index; CDEIS: Crohn’s disease endoscopy index of severity; CRP: C-reactive protein; CTE: computed tomography enterography; FAP: fibroblast activation protein; FAPi: fibroblast activation protein inhibitor; FCP: fecal calprotectin; FDG: fluorodeoxyglucose; MTI: magnetization transfer magnetic resonance imaging; PET/CT: positron emission tomography/computed tomography; SES-CD: simple endoscopic score for Crohn’s disease.

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
