# Peer review of "FAPi PET/CT Imaging to Identify Fibrosis in Immune-Mediated Inflammatory Diseases"

_biomedicines, 2025, doi:10.3390/biomedicines13040775_

Round 1
Reviewer 1 Report
Comments and Suggestions for Authors
The manuscript is very well written and inclusive review.
In this review article authors reviewed the published studies which investigated the FAPi PET/CT in diagnosing, surveillance and treatment of various diseases, in a very well written and well organized matter.
Using FAPi PET is a new technique recently being used in multiple clinical trials and in clinical practice and is of high importance. This review briefly reviewed this methods efficacy in various fields of medicine. I would suggests acceptance of this article.
Reviewer 2 Report
Comments and Suggestions for Authors
Comment
This review by Löwenberg provides an overview of the use of FAPi PET/CT imaging to visualize fibrosis in various immune-mediated inflammatory diseases (IMIDs). It highlights the high sensitivity of FAPi PET/CT in detecting early fibrosis and its correlation with disease severity across different IMIDs, emphasizing the superiority of FAPi PET/CT over conventional imaging modalities for fibrosis assessment. Additionally, the review discusses the potential of FAPi PET/CT as a tool for evaluating disease severity, staging, and monitoring fibrosis-related organ damage. Finally, the review calls for further studies and optimized imaging protocols to further validate its diagnostic value in a broader range of IMIDs.
The paper could be published in Biomedicines after addressing the following revisions:
1 I see Table 2 and 3 in the review. So where is Table 1?
2 Please separate Table 2 into different Table and put them in different parts (e.g., Pulmonary diseases, IgG4-related diseases, IgG4-related diseases, and Renal diseases). Please do not simply put the data together and make it so large.
3 Please provide a justified manuscript instead of the left alignment version.
4 For a review paper, I suggest you add some figures in different parts and give detailed descriptions of the figures. I am surprised to find that there is no figure in a review.
5 In ‘6. Gastro-intestinal diseases’ part, you put liver and gastrointestinal diseases together. I don’t think this is a good idea. Please separate it into 2 parts or revise the title of Part 6.
6 The author uses IgG4-related diseases as the sub-title of part 3. But it is not consistent with the other sub-titles in different parts because they are all related with organ diseases. So my suggestion here is that you can also use organ-related diseases as the sub-title (e.g., Glandular-related diseases).
Reviewer 3 Report
Comments and Suggestions for Authors
This review aimed to clarify the role of FAPi PET/CT imaging in detecting fibrosis in immune-mediated inflammatory diseases (IMIDs). The authors have included a wide range of recent studies, providing a clear and up-to-date analysis. By reviewing studies on the use of FAPi PET/CT in detecting fibrosis across various systemic diseases, we have gained a relatively comprehensive understanding of the advancements in this field. The structure and logic of the article are clear and well-organized.
However, I have a question regarding the scope of diseases mentioned in the review. To my knowledge, inflammatory joint or skin diseases are common types of IMIDs. What is the current state of research on the application of FAPi PET/CT in these conditions? Perhaps the authors could provide a more comprehensive discussion on the use of FAPi PET/CT in these common IMIDs, even if it turns out that there is limited research in this area.
Round 2
Reviewer 2 Report
Comments and Suggestions for Authors
Accept in present form
Reviewer 3 Report
Comments and Suggestions for Authors
I have reviewed the revised version of the manuscript and the authors' reply to my previous review comments. The authors have made significant improvements to the manuscript, particularly in the display of the tables. I think the manuscript has been improved and now meets the standards required for publication. I believe this article will be of interest to the readers and will contribute meaningfully to the progress in improving the understanding of the role of FAPi PET/CT imaging in detecting fibrosis in immune-mediated inflammatory diseases (IMIDs).